# Interfacial Bond Properties of Underwater Concrete Coated with Bisphenol A Epoxy Resins

**DOI:** 10.3390/polym15214290

**Published:** 2023-11-01

**Authors:** Sungwon Kim, Jin-Hak Yi, Hyemin Hong, Seoung Ik Choi, Dongchan Kim, Min Ook Kim

**Affiliations:** 1Ocean Space Development and Energy Research Department, Korea Institute of Ocean Science and Technology, 385 Haeyang-ro, Yeongdo-gu, Busan 49111, Republic of Korea; swkim@kiost.ac.kr (S.K.); yijh@kiost.ac.kr (J.-H.Y.); hyeminhong@kiost.ac.kr (H.H.); sichoi@kiost.ac.kr (S.I.C.); 2Department of Civil Engineering, Seoul National University of Science and Technology, 232 Gongneung-ro, Nowon-gu, Seoul 01811, Republic of Korea; kdc0357@seoultech.ac.kr

**Keywords:** epoxy coating, interfacial properties, marine environment, exposure duration, bond strength, concrete substrate

## Abstract

This study investigated changes in the interfacial properties of epoxy-coated concrete exposed to various conditions, regarding the epoxy type, coating equipment, and exposure environment and period. The measured coating thickness and pull-off bond strength exhibited diverse trends, depending on the exposure period and conditions. In the real sea (RS) environment, the average bond strengths for bisphenol A (BPA) (E1), BPA with zinc powder (E2), and BPA with cresyl glycidyl ether (E3) were 1.26, 1.93, and 1.92 MPa, respectively. The coating method did not significantly affect the measured coating thickness and strength values. The conventional roller (D1) exhibited the highest thickness variation, with a value of 214.45 μm. The RS condition significantly increased the coating thickness (34% to 158%) compared to the tap water (TW) condition. The exposure conditions had little impact on bond strength except for E3, which showed an increased strength (2.71 MPa) over 7–91 days, especially under RS conditions, while E2 remained constant at approximately 1.82 MPa. This study offers insights into factors influencing marine concrete coating performance and discusses limitations and future work.

## 1. Introduction

Marine concrete structures face exposure to harsh and aggressive environments, such as chloride and sulfate attacks, which can significantly impact their service life. Underwater coatings are recognized as an economical and rapid repair method for preventing water and moisture infiltration into the concrete. Numerous studies have been conducted to develop effective repair strategies for concrete structures with surface protection, with some focusing on specific coating types [1,2,3,4,5,6,7,8,9,10,11]. For instance, Pan et al. [1] noted that applying a coating to a concrete surface forms a protective polymer film that hinders the penetration of corrosive substances, thus reducing structural damage and extending the structure’s lifespan. Diamanti et al. [2] conducted an experimental study on coatings, finding that polymers in coatings decrease porosity and enhance protection. Epoxy, a well-investigated coating material, effectively shields concrete from harmful ions such as chlorides and sulfates. Almusallam et al. [3] reported that epoxy and polyurethane coatings offer superior adhesion and durability compared to acrylic, chlorinated rubber, and polymer emulsion coatings. Furthermore, Almusallam et al. [4] exposed concrete specimens coated with epoxy and polyurethane to a 2.5% sulfuric acid solution for 60 d. The results showed partial damage at the specimen edges, while the remaining sections remained relatively intact. Aguirre-Guerrero et al. [5] found epoxy-based coatings to be the most effective for safeguarding concrete, based on water permeability and chloride penetration resistance test results. Kang et al. [6] discussed enhancing the friction and wear resistance of epoxy resins by incorporating Nano-SiO_2_. Hang et al. [7] described the effectiveness of pretreatment with a precoating solution within a 20–40% concentration range, which improved the interfacial bond performance between the repair material and substrate by filling irregular defects on the surface. Chruściel et al. [8] reported epoxy resins as highly practical polymers and suggested that their properties can be improved by adding reactive silanes, polysiloxanes, silica, and other substances. Moradllo et al. [9] conducted a five-year study on the durability of concrete coated with six different types of coatings placed in a tidal zone. They found that acrylic-based and epoxy-polyurethane hybrid resin coatings effectively prevented chloride ion penetration. Won et al. [10] investigated the impact of water exposure on the interfacial bond between an applied epoxy coating and a concrete substrate, concluding that the coating thickness was not directly related to the measured bond strength. Abdelkader et al. [11] reported that the type of hardener applied to epoxy resin significantly influenced moisture absorption and epoxy performance. They also found that the type of hardener used affected the post-performance. Prior research has consistently noted the prevalent use of polymer-based materials in concrete coatings. These coatings have been assessed in terms of their ability to enhance the durability of the intended concrete structures and improve the interfacial bond properties between the coating and the concrete. Notably, epoxy and polyurethane have emerged as the top-performing materials among polymer-based coatings. It is important to highlight that, even within the same polymer category, variations in bond strength have been observed depending on the manufacturer. Various epoxy resins, including BPA, Bisphenol F (BPF), novolac, elastomeric, and zinc-rich epoxy resins, have found application in underwater coating [12,13,14,15,16]. Among these, BPA is recognized for its enhanced adhesion and chemical resistance, but it does have a vulnerability to ultraviolet (UV) degradation, leading to potential brittleness [17]. Furthermore, BPA offers cost-effectiveness in comparison to novolac epoxy resin and ease of application when contrasted with zinc-rich epoxy resin [18,19].

The coating method, encompassing coating tools, the coating procedure, and the type of pre-treatment, plays a significant role in shaping the interfacial bond properties between the coating and the substrate. Sadowski et al. [20] affirmed that the maximum aggregate size has an impact on the interfacial properties between epoxy resin and the concrete substrate. Kim et al. [21,22] introduced coating devices originally designed for underwater coating of marine structures and conducted a performance evaluation, comparing them with conventional rollers. They concluded that the conventional roller outperformed the new devices, owing to its ability to ensure a more uniform distribution of the coating, with a reduced coating thickness.

In their respective studies, Al-Kheetan et al. [23] examined the adhesion of polymer-coated concrete over increasing underwater exposure durations, noting a rise in adhesive strength from 0.42 MPa after 24 h to 1.66 MPa after 408 h. Sadati et al. [24] reported a substantial reduction in carbonation depth, from 4.0 mm in uncoated concrete specimens to around 1.0 mm in specimens coated with epoxy-polyurethane after an 88-month exposure to marine and soil environments. Basheer et al. [25] emphasized the need for further investigation into the long-term performance of surface coatings, despite noting improvements in short-term structural performance. The exposure environment was identified as a crucial factor influencing coating performance. Kim et al. [26] highlighted reduced pull-off bond strength during underwater coating applications compared to ambient conditions. Dolan et al. [27] stressed the significance of research into underwater coatings owing to their substantial impact on epoxy performance in underwater versus atmospheric environments. Lastly, Toutanji et al. [28] explored polyurethane coating technology, demonstrating its ability to enhance concrete’s structural performance, including bending and compression, even in seawater environments.

This study experimentally investigated the significant factors, including the epoxy resin type, coating equipment, exposure time, and exposure conditions, that must be considered when applying coatings to marine concrete structures. This study considered three different epoxy resins and coating equipment options, based on the results of previous studies. In addition, the effects of the exposure period and conditions on the measured pull-off bond were evaluated. The investigated test variables, materials, and experimental procedures are summarized in the following sections.

## 2. Research Significance

The majority of the prior research on concrete surface repair has primarily concentrated on conventional, land-based structures, with less emphasis on marine structures. This discrepancy can be ascribed to the limited understanding of underwater coating materials, their suitability for real marine environments, and the reliability of experimental outcomes. Experimental studies that specifically addressed the bond performance of underwater coatings in real sea conditions are currently limited in the literature. Such research would hold significant value for individuals engaged in the repair and retrofitting of submerged concrete structures. In addition, studies on the development and performance evaluation of coating materials for underwater operations are limited, and the time-dependent interfacial adhesion between applied coatings and substrates has not been systematically investigated. Therefore, this study performed experiments to determine the bond strength between underwater coatings and concrete substrates. The results presented in this paper are part of an ongoing research project and are expected to greatly contribute to the maintenance of marine concrete structures located in actual marine environments.

## 3. Experimental

### 3.1. Materials and Test Variables

Figure 1 summarizes the test variables used in this study. As depicted in Table 1, this study took into account three distinct types of epoxy resin and one type of hardener, guided by previous research findings [10,21,22,26]. All of the epoxy resins are readily available commercially and are suitable for use in underwater conditions. The nomenclature for each coating is set as E1 (RS 500P, Chemco, Coatbridge, Scotland, UK), E2 (Alocit 28.14, A&E Group, Shah Alam, Malaysia), and E3 (Alocit 28.15, A&E Group, Shah Alam, Malaysia) throughout the study. Specifically, the E1 epoxy resin is bisphenol A (BPA), which can be used in various applications, including coatings, civil engineering, adhesives, and electrical materials for insulation purposes. The E2 epoxy resin, comprising BPA with zinc powder, was the subject of investigation in this study, owing to prior research findings indicating its superior corrosion and scratch resistance [29]. The size of zinc particles ranged between 100 and 200 µm. The zinc powder included in E2 has a percent composition ranging from 50% to 60%, and it is of a mono-constituent type. Finally, BPA with cresyl glycidyl ether (BGEI) is denoted as E3. BGEI is known to exhibit excellent adhesion underwater and in air, as well as resistance to mechanical wear and chemical substances. Table 2 presents the ion concentrations in seawater and tap water, excluding extremely small values. The actual marine exposure test was conducted near Nambu-myeon, Geoje-si, Gyeongsangnam-do, Republic of Korea, considering periodic measurements, monitoring, and safety conditions. Table 3 presents the pH, salinity, and temperature data for seawater employed in both the laboratory and real sea experiments. It was established that there were no significant disparities in pH and salinity between the seawater in the laboratory and in the real sea environment. Figure 2a,b depict the fluctuations in ambient air and seawater temperatures, respectively, recorded at the real sea test site over the course of the testing period. While the laboratory tests employed controlled conditions using tap water or seawater, a notable variation in temperature was observed during the real sea (RS) testing period, as illustrated in Figure 2b. As shown in Figure 3a–c, three distinct coating tools were selected for underwater coating: a conventional roller (D1) and two coating devices (D2 and D3). The D2 and D3 devices were constructed from polytetrafluoroethylene (PTFE), to enhance their lightweight properties and corrosion resistance. Teflon fibers were used as the brush material, to ensure uniform distribution of each coating. D2 used a piston to inject and push out each coating material. When the valve is opened, the piston moves toward the nozzle, due to the stiffness of the spring, allowing the coating material to be sprayed onto the brush, facilitating the coating process. The D3 equipment comprised a coating material storage tank, an oxygen tank, and a brush with a gauge attached, used to monitor air pressure changes. Figure 3d–f show photographs of the coating process using each device in an actual marine environment. The coated concrete specimens were collected on predetermined dates with exposure periods of 7, 28, 56, and 91 d, and tests were subsequently performed. The effects of tap water and seawater on the interfacial properties between the coating and concrete substrate were investigated in a laboratory under controlled temperature and humidity conditions. In the following section, the nomenclature shown in Figure 1 is used to simplify the test analysis. For example, E3D3-RS-28D refers to a concrete sample coated with E3 using D3 and exposed to RS for 28 d.

### 3.2. Concrete Specimen Preparation

A 300 mm × 300 mm × 30 mm rectangular concrete substrate was prepared. A mixer truck and electric concrete vibrator were used to cast the concrete. The adopted water-to-cement ratio (w/c) was 0.24, and ordinary Portland cement (OPC) was used. The prepared specimens were water-cured for 28 d under controlled temperature and humidity conditions. Subsequently, the surface was water-jetted and relocated to the laboratory or onsite for surface coating. The compressive and splitting tensile strengths of the substrate concrete were measured according to the ASTM C39 and ASTM C496, respectively [30,31]. At least four cylindrical specimens (D100 mm × H200 mm) were used to determine the compressive and splitting tensile strengths. The averaged 28 d compressive and splitting tensile strengths were 75.69 MPa and 4.08 MPa, respectively.

### 3.3. Test and Data Analysis

After establishing the appropriate test duration for each exposure condition, the coated specimens were extracted. The specimens were then initially cut into small rectangular sizes of 20 mm × 20 mm × 20 mm to capture scanning electron microscopy (SEM) images and observe the changes at the interface between the coating and substrate. These images were captured using an EM-30AX device manufactured by COXEM, Daejeon, Republic of Korea, following the completion of the proper curing, grinding, and polishing processes. Images were captured at two exposure periods of 7 and 91 d, and two different magnifications (×100 and ×1000) were used at the same location. The presence of separations, residual voids, and cracks near the interface was visually inspected to investigate the interfacial bond behavior. The coating thickness was measured according to the ASTM D6132 [32]. An Elcometer A500C-B (Elcometer, Manchester, England, UK) was used for the thickness measurements. Subsequently, a pull-off bond test was conducted in accordance with the ASTM C1583 procedure [33]. A commercially available pull-off bond test device (Elcometer F510-50S, Elcometer, Manchester, UK) was used for the measurements. The specimen surface was cleaned before attaching it to the dolly using epoxy glue, and the diameter of the dolly used was 50 mm. After attaching the dolly, the specimens were stored in the laboratory for more than 24 h to ensure proper adhesion. A pull-off rate of 0.02 MPa/s was used to prevent unexpected failure. The thickness was determined nondestructively, and the bond strength was estimated by dividing the measured maximum force (N) by the contact area (mm^2^). One-way analysis of variance (ANOVA) and Tukey’s honest significant difference (HSD) tests were used for the statistical analysis of the measured values to determine whether there were statistically significant differences in the means of the results. The null hypothesis for the ANOVA was that each group would have similar values and not exhibit significant differences. In contrast, the HSD test was used to detect differences between any pair of groups.

## 4. Test Results

### 4.1. Measured Thickness

The coating thickness was measured according to the ASTM D6132 procedure; Table 4 lists the measured values. The maximum and minimum values were 635.56 µm (E3D3-RS-28D) with a standard deviation (S.D.) of 99.23 µm and 27.78 µm (E1D1-TW-7D) with a S.D. of 5.17 µm, respectively. The values displayed substantial variations, irrespective of the test variables. The high variability in underwater coating thickness could have been the result of the complex interplay between environmental factors, application challenges, material properties, and unique conditions of the underwater environment [34,35,36,37]. Thus, achieving a consistent and uniform coating thickness under such conditions requires careful planning, specialized techniques, and an understanding of the factors contributing to these variations. The range of the measured values fell within that of a previous study, which reported a range between 0.2 mm and 0.95 mm [10]. The following sections present detailed comparisons and analyses.

### 4.2. Measured Pull-Off Bond Strength

The measured strengths varied significantly, as shown in Table 4. The maximum and minimum strengths measured under real sea (RS) conditions were 3.21 MPa (E3D3-RS-28D) with an S.D. of 0.18 MPa and 0.09 MPa (E2D2-RS-91D) with an S.D. of 0.03 MPa, respectively. Nonetheless, it was anticipated that the bond strength measured under laboratory conditions using tap water would demonstrate a higher strength value compared to the other conditions. This expectation was rooted in our prior research, which indicated that the pull-off bond strength measured under consistent temperature and humidity conditions consistently yielded higher results when contrasted with measurements taken under real sea (RS) conditions [10,21,22]. The exact causes behind this notable onsite variability remain undetermined, but they could be linked to uncontrolled variables and unforeseen factors, including temperature fluctuations, humidity levels, tidal effects, and the presence of marine organisms. The correlation coefficient confirmed a weak relationship between the average thickness and bond strength. The coating device used may significantly influence the thickness, whereas the exposure conditions and duration could more likely affect the bond strength. The estimated correlation coefficient of 0.11 was insignificant, and these findings have implications for underwater coatings. Hence, it is advisable for workers not to excessively apply coating material for underwater applications. Importantly, these findings align with those from a prior study conducted in controlled laboratory conditions using different equipment [10].

## 5. Analysis

### 5.1. Effect of Epoxy Type on the Measured Thickness and Bond Strength

The mean thickness values for E1, E2, and E3 were 117.54, 204.60, and 227.40 µm, respectively, as shown in Figure 4a. The ANOVA results indicated a significant difference in the mean coating thickness values among the three coatings, with a *p*-value of less than 0.0001 (Table 5). Therefore, the variation in the coating thickness can be confidently attributed to factors other than chance. Post hoc analysis conducted using Tukey’s HSD test indicated significant differences in the mean coating thickness values between E1 and E2 (*p* = 0.0001) and between E1 and E3 (*p* = 0.0001). However, no significant difference was observed in the mean coating thickness between E2 and E3 (*p* = 0.3222). The mean bond strengths of E1, E2, and E3 were 1.26, 1.93, and 1.92 MPa, respectively, as shown in Figure 4b. The ANOVA results highlighted a significant difference between the mean bond strengths of the three coatings (*p* < 0.0001), as shown in Table 5. Thus, the disparity in the mean bond strengths can be confidently attributed to factors beyond chance. The Tukey’s HSD test revealed that E2 exhibited a higher mean bond strength than E1 and E3. Similar trends were observed when the test results were compared with those of previous studies [10]. E1 displayed the lowest mean bond strength of 1.26 MPa and a mean thickness of 117.54 μm, while the remaining two coatings displayed similar average strengths and thicknesses. This suggests that factors other than the type of epoxy exerted a more substantial influence on both the strength and thickness. It can also be seen that the two additives, zinc powder and cresyl glycidyl ether (CGE), made a difference in performance among the three coatings. Zinc powder tends to increase viscosity by physically occupying space within the epoxy matrix, leading to a thicker consistency. Conversely, CGE may not directly elevate viscosity in the manner of zinc powder. Instead, it may have an impact on the curing process and the ultimate properties of the epoxy, potentially affecting its flow characteristics.

SEM was used to investigate the internal porosity within each epoxy coating, and the results from SEM under the RS condition confirmed that E1 had larger internal pore sizes than E2 and E3, as shown in Figure 5a–f. The presence of these pores could have resulted from seawater intrusion during underwater coating, potentially affecting the interfacial bond strength, as previously reported [38,39]. Pores and microcracks are known to have a negative impact on interfacial bond strength. Therefore, when selecting coating materials for underwater applications, it is essential to consider the size and distribution of pores. Although E1 exhibited larger and more pores, as shown in Figure 5, this research did not systematically investigate the size of pores and its effects on coating performance. The degree of porosity is a critical factor influencing a coating’s overall effectiveness in providing corrosion resistance, adhesion, and long-term durability. This aspect should be taken into consideration in future research related to coating applications.

### 5.2. Effect of the Coating Method on the Measured Thickness and Bond Strength

Figure 6a shows the mean thickness values of 182.29, 175.28, and 171.88 µm for D1, D2, and D3, respectively. The values exhibited a high variability, whereas the average values were similar. The ANOVA indicated no significant differences in the mean thicknesses according to the method used (see Table 6). Specifically, the F-statistic was 0.239, which was insignificant at a 0.05 significance level, indicating no statistically significant difference between the mean thicknesses of the three methods. The *p*-value of 0.787 further supported this conclusion. Notably, D1 exhibited the highest variation with an S.D. of 123 µm, which was unexpected considering that D1 was initially anticipated to have a stable thickness compared with other methods based on previous findings [10,21,22]. The mean bond strengths of D1, D2, and D3 were 1.87, 1.70, and 1.58 MPa, respectively, and no significant difference between the groups was observed at the 0.05 significance level. Overall, the effects of the coating method on the measured thicknesses and strengths were insignificant. Generally, the coating method used can significantly influence the interfacial bond properties. The values measured in this study confirmed that measurements can vary considerably owing to environmental conditions and human error. Repair work in underwater conditions can disrupt visibility, owing to turbidity and the presence of particulate matter [40,41,42]. Thus, further studies are necessary regarding the coating method, to ensure a proper interfacial bond between the coating and target structures.

### 5.3. Effect of Exposure Condition on the Measured Thickness and Bond Strength

Figure 7a,b illustrates the effects of the exposure conditions on the thickness, depending on the epoxy type and coating method, respectively. As anticipated, the thickness with the highest S.D. was observed under the RS conditions. This implies that the thickness values under RS conditions were more dispersed than those under the other conditions. A slight trend was observed, wherein the thickness increased from TW or SW to RS, regardless of the type of epoxy and the coating method used. Statistical analysis was conducted to confirm the significant differences between E1 and D3 with changing exposure conditions at the 0.05 significance level. Figure 7c,d shows changes in the bond strength values depending on the exposure conditions. E1 and E2 exhibited relatively stable strength values compared with E3, regardless of the exposure conditions. The ANOVA yielded that the *p*-values for E1 and E2 were greater than 0.05, suggesting no significant changes in the strength values. These results also reaffirmed the insignificant correlation between strength and thickness. Furthermore, the bond strength values were found to be similar under all conditions at the 0.05 significance level, with the coating method having no significant effect. Overall, it can be inferred that the exposure condition can significantly influence the coating thickness, resulting in potentially thicker and more dispersed coatings compared with conventional conditions. Furthermore, within the range of this study, the exposure conditions had a limited impact on the measured bond strength, except for E3.

### 5.4. Effect of Exposure Periods on the Measured Thickness and Bond Strength

Figure 8a,b show the changes in the mean thickness values for each material with increasing exposure periods. The thickness values remained relatively constant in the TW and SW conditions, regardless of the exposure period, whereas the values decreased in the RS conditions. Initially, the thickness was to be determined at the time of initial application, and no significant change over time was expected. This expectation was confirmed under constant laboratory conditions, and the effect of the coating method on changes in coating thickness over the exposure period was relatively minimal. The decreasing trend in thickness observed only in the RS conditions can be attributed to surface abrasion issues; nonetheless, further investigation remains necessary, and nondestructive methods could be useful for determining the reason for this reduction [43,44,45,46]. However, it should be noted that the thickness did not exhibit a significant relationship with the strength within the range of this study.

Neither E1 nor E2 exhibited significant changes or trends in strength during the exposure period (Figure 8c,d). By contrast, E3 exhibited a slight trend of increasing bond strength with increasing exposure time, particularly under the RS condition. The SEM results indicated that E3 exhibited a stronger and thicker interfacial bond compared with that of E1 and E2 (Figure 9).

E3 exhibited a slight increase in bond strength with an increased exposure period, especially in the RS condition, while E1 and E2 showed constant values. The two additives, zinc powder and cresyl glycidyl ether (CGE), resulted in performance variations among the three coatings. As mentioned previously, while zinc powder tends to increase viscosity and lead to a thicker consistency, CGE may not directly increase viscosity like zinc powder. However, it can influence the curing process and the final properties of the epoxy, potentially affecting its flow characteristics. These results are consistent with those of Park et al. [47], who reported an improved interfacial bond using E3. The improved bonding properties may be related to the incorporation of a low-viscosity reactive diluent (cresyl glycidyl ether), which may positively impact the mechanical properties and corrosion resistance. Initially, the bond strength was expected to gradually increase in the TW condition, owing to the consistent temperature and humidity compared with that in the RS condition. The effect of the coating method on the changes in the bond strength with exposure time was minimal, and the epoxy type more significantly influenced the interfacial bond than the coating method.

Overall, this study confirmed that the exposure period and conditions had a combined effect on the early strength gain, suggesting that further long-term investigations are necessary.

## 6. Discussion

This study aimed to establish and conduct an experimental program to investigate the significant factors influencing the surface protection of marine concrete structures. The coating performance was assessed by measuring both the coating thickness and bond strength. It was observed that the exposure conditions and period significantly influenced the coating thickness and strength. The results also indicated a high degree of variability. However, it is imperative to emphasize the need for further research in underwater coatings for marine structures. These structures are constantly exposed to harsh underwater environments, which can result in material corrosion and degradation over time. Continuous research efforts can play a pivotal role in the development of and improvement in coatings that offer effective corrosion protection, thereby extending the lifespan of these structures and ultimately reducing maintenance costs [48,49,50,51,52,53]. The environmental impact of underwater coatings is a major concern [54,55,56]. Further research is essential, to develop environmentally friendly coatings that release less harmful chemicals into water, preserve marine ecosystems, and comply with environmental regulations. As discussed previously, marine conditions can vary significantly owing to various factors, such as temperature, salinity, and water quality. Coatings can be formulated to meet the specific requirements of each location or structure, while maintaining effective performance under various conditions. Research is also necessary to develop cost-effective coating solutions that provide long-term protection by reducing the maintenance and operating costs of marine structures [57,58].

In this study, we noted significant variability in the test results, primarily stemming from uncontrolled conditions and human errors. To mitigate this variability in underwater coatings and ensure reliability, a multifaceted approach is essential. This encompasses factors such as meticulous material selection, precise application techniques, stringent quality control measures, and continuous maintenance. In particular, the proper surface preparation of target structures and thorough training for applicators play pivotal roles in achieving the desired performance. Regular monitoring and maintenance are equally vital to secure long-term effectiveness. The high variability in the results can be considered a limitation of this study, as it may hinder the proper selection of coating materials and methods, considering their appropriate application areas. Similarly, reducing the high variability in underwater coatings and achieving reliability involves factors such as proper material selection, application techniques, quality control measures, and ongoing maintenance. These limitations should be properly addressed through ongoing testing and future research.

Within this research, it was further affirmed that the exposure conditions had the most substantial influence on both the thickness and strength, while the effect of the coating method was deemed negligible following a thorough statistical analysis. Specifically, the differences between the maximum and minimum mean thickness and strength values were 83% and 37% higher in the RS condition compared to the TW condition, while no significant difference in the coating method was observed at the 0.05 significance level. This contradicts the results obtained in previous laboratory testing. Hence, it is evident that conducting tests in authentic sea conditions is indispensable. This is because the performance of underwater coatings on concrete structures can exhibit substantial variations between controlled laboratory settings and the real sea environment. In addition, within the scope of this study, E3 can be recommended to ensure proper bonding performance for underwater structures, while the coating method could not be optimized.

Additional research is required to gain a deeper understanding of the wear resistance and fatigue characteristics of underwater coatings through long-term investigations. Conducting extended studies is essential to enhance the performance assessment of underwater coatings. It is worth noting that short-term studies may not provide a precise reflection of the long-term performance of these coatings. Over time, various environmental factors, such as saltwater corrosion, biofouling, and other wear and tear, can significantly affect the effectiveness of these coatings. Long-term studies provide a more realistic simulation of the actual conditions these coatings will face in practice, while allowing for a comprehensive assessment of the durability and robustness of underwater coatings. This encompasses the evaluation of their resistance to deterioration, the preservation of their protective properties, and their ability to endure prolonged exposure to the demanding conditions underwater. Extended testing proves invaluable in uncovering any latent weaknesses or vulnerabilities in coatings that might escape notice in shorter-term investigations. This, in turn, leads to refinements and enhancements in the formulation and application of these coatings, ultimately bolstering their long-term performance. In summary, continuous research into underwater coating technologies is imperative, to tackle the distinct challenges presented by marine environments, extend the longevity and efficacy of marine structures, and adhere to evolving environmental and regulatory standards, ensuring that these coatings retain their effectiveness, safety, and environmental sustainability over time.

## 7. Conclusions

This study conducted experimental investigations into the alterations in the interfacial bond properties of BPA epoxy resins when applied to underwater concrete. It specifically examined the impact of various epoxy types, coating techniques, and exposure conditions and durations on both the coating thickness and bond strength. The following conclusions have been derived from the test results and subsequent analyses:BPA (E1, control) had the lowest coating thickness and bond strength, whereas BPA with zinc powder (E2) and BPA with cresyl glycidyl ether (E3) demonstrated similar values.The impact of the chosen coating method on the measured thickness and strength values appeared to be minimal, and notably, the conventional roller (D1) had the highest thickness variation. This might be attributed to the possibility that the other factors had a more significant influence than the coating method.Under RS conditions, there was a noticeable increase in coating thickness compared to the controlled condition. As for bond strength, the exposure conditions had only a limited effect, except for E3.Thickness values remained relatively stable regardless of the exposure period, and E3 demonstrated an upward trend in bond strength with increasing exposure time, particularly under the RS conditions.The relationship between the measured thickness and bond strength appeared unclear, and this pattern aligns with findings from previous studies.

The measured thickness and strength values exhibited high variability, which could be attributed to environmental factors (e.g., uncontrolled temperature, humidity, and the presence of tides). This study offers valuable insights into the key factors that need to be considered when applying coatings to marine concrete structures, thereby contributing to the ongoing research and development of underwater coating technologies.

## Figures and Tables

**Figure 1 polymers-15-04290-f001:**
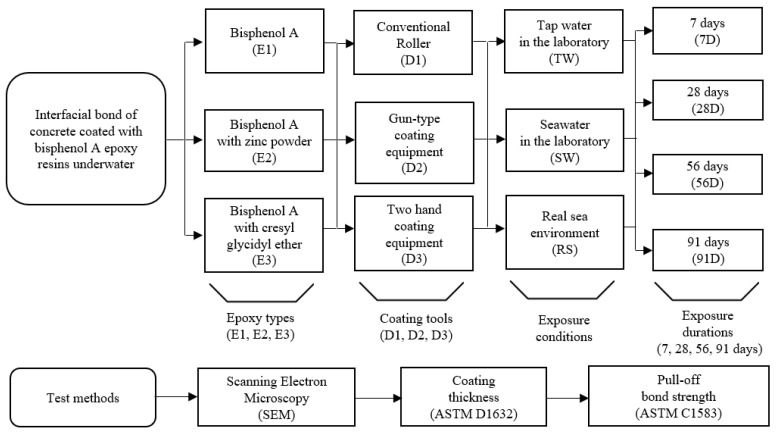
Test variables and test methods.

**Figure 2 polymers-15-04290-f002:**
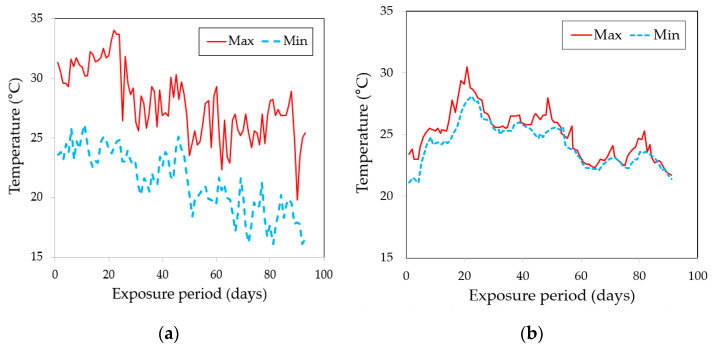
Temperature changes during the test period in (**a**) ambient air and (**b**) seawater.

**Figure 3 polymers-15-04290-f003:**
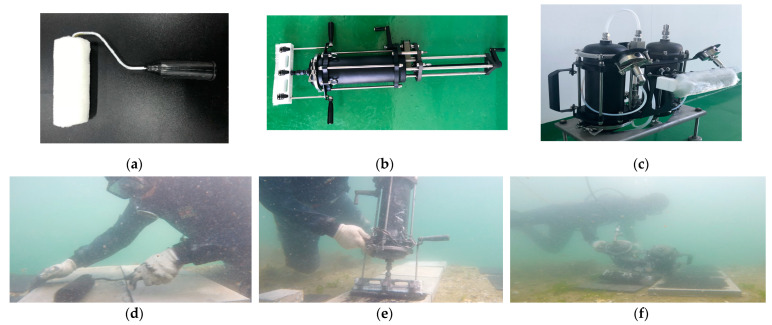
Photos of the three different coating methods considered in this study and their application under real sea conditions: (**a**) D1–Roller; (**b**) D2–Gun type; (**c**) D3–Two-handed type; (**d**) application using D1; (**e**) application using D2; (**f**) application using D3.

**Figure 4 polymers-15-04290-f004:**
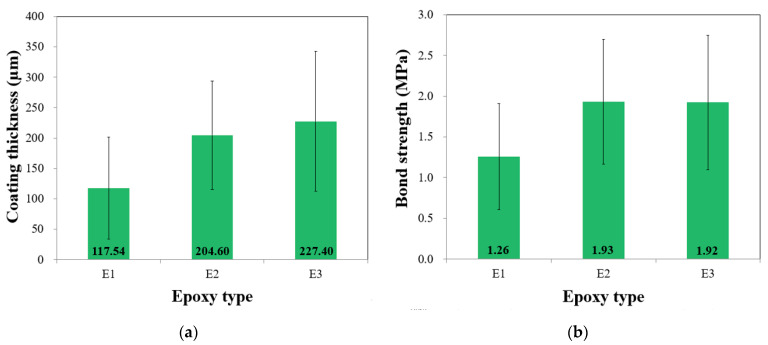
Effect of epoxy type on (**a**) coating thickness and (**b**) bond strength.

**Figure 5 polymers-15-04290-f005:**
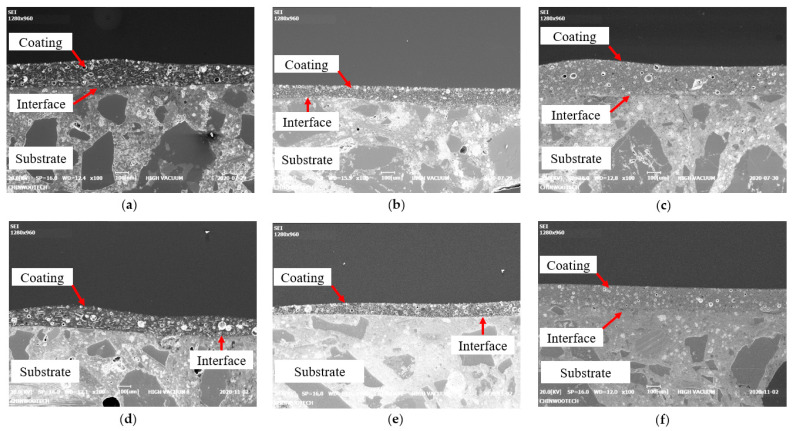
SEM images (at ×100 magnification) based on type of epoxy and exposure period, with constant coating method (D2) and exposure condition (RS): (**a**) E1D2-RS-7D; (**b**) E2D2-RS-7D; (**c**) E3D2-RS-7D; (**d**) E1D2-RS-91D; (**e**) E2D2-RS-91D; (**f**) E3D2-RS-91D.

**Figure 6 polymers-15-04290-f006:**
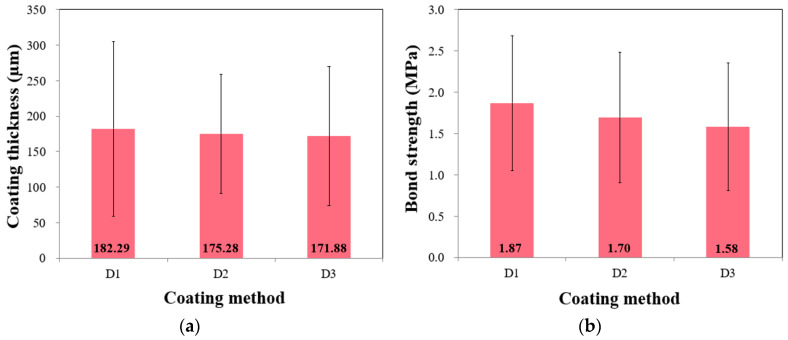
Effect of coating method on (**a**) coating thickness and (**b**) bond strength.

**Figure 7 polymers-15-04290-f007:**
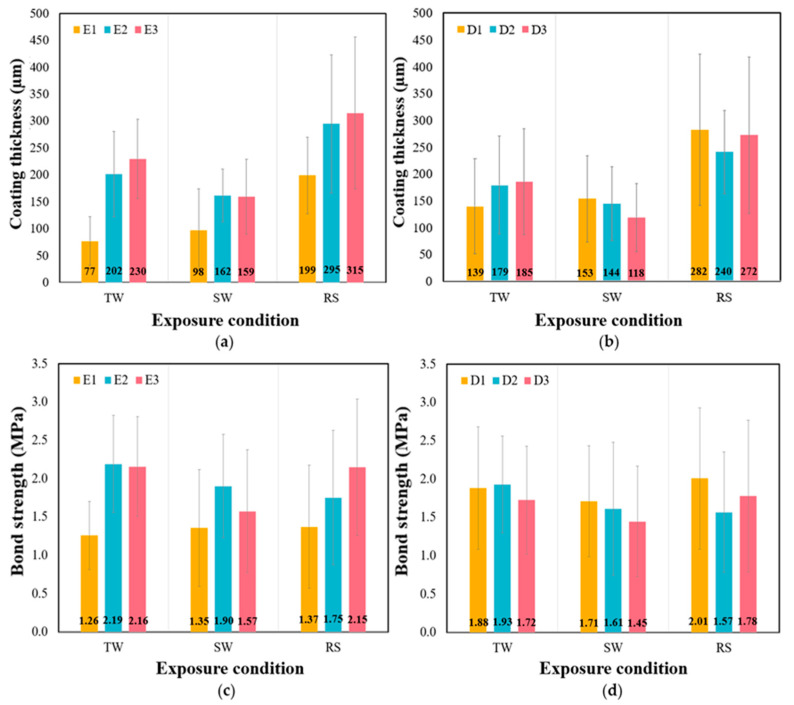
Effect of exposure condition on (**a**) coating thickness according to the type of epoxy; (**b**) coating thickness according to coating method; (**c**) bond strength according to the type of epoxy; (**d**) bond strength according to coating method.

**Figure 8 polymers-15-04290-f008:**
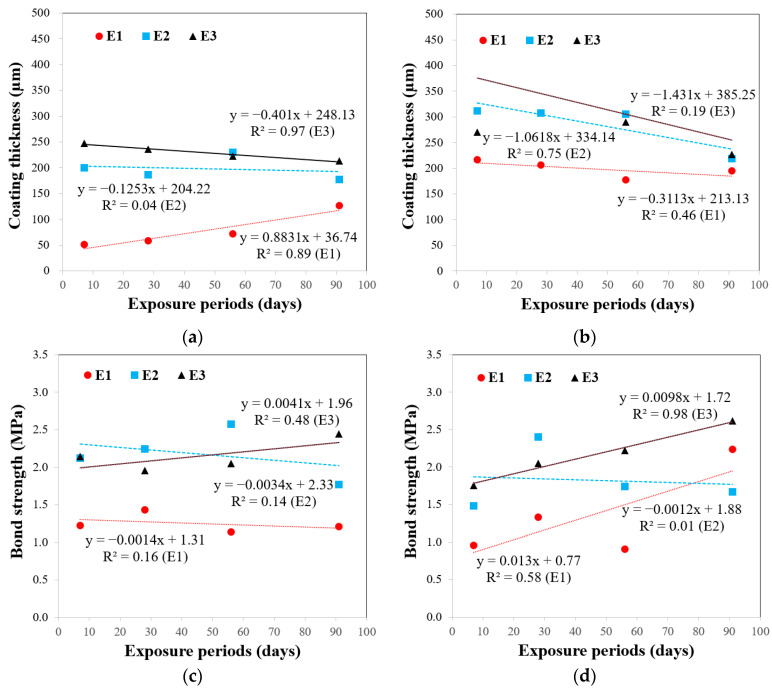
Effect of exposure period on (**a**) coating thickness under TW condition; (**b**) coating thickness under RS condition; (**c**) bond strength under TW condition; (**d**) bond strength under RS condition.

**Figure 9 polymers-15-04290-f009:**
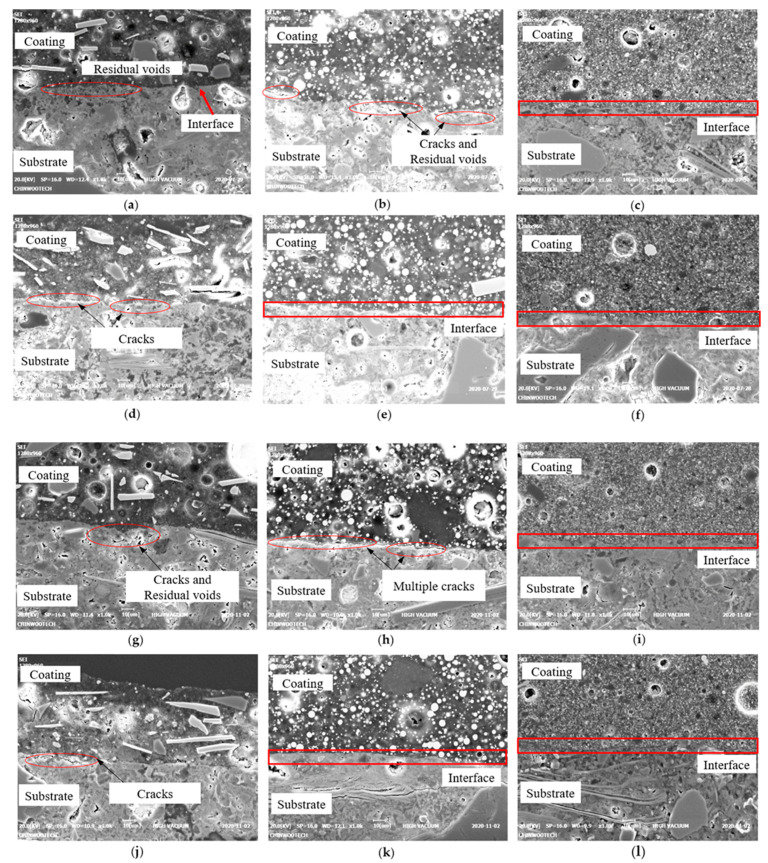
SEM images (at ×1000 magnification) based on type of epoxy, coating method, and exposure period, with a constant exposure condition (RS): (**a**) E1D2-RS-7D; (**b**) E2D2-RS-7D; (**c**) E3D2-RS-7D; (**d**) E1D3-RS-7D; (**e**) E2D3-RS-7D; (**f**) E3D3-RS-7D; (**g**) E1D2-RS-91D; (**h**) E2D2-RS-91D; (**i**) E3D2-RS-91D; (**j**) E1D3-RS-91D; (**k**) E2D3-RS-91D; (**l**) E3D3-RS-91D.

**Table 1 polymers-15-04290-t001:** Type of epoxy resin.

Nomenclature	Epoxy Resin	Hardener	Mixing Ratio (Resin:Hardener)	Density (g/cm^3^)	Product
E1	Bisphenol A (C_15_H_16_O_2_)	Isophorone diamine (C_10_H_22_N_2_)	5.1:1.0	1.60	RS500P
E2	Bisphenol A with zinc powder	Isophorone diamine (C_10_H_22_N_2_)	5.0:1.0	1.82	Alocit 28.14
E3	Bisphenol A with cresyl glycidyl ether (C_10_H_12_O_2_)	Isophorone diamine (C_10_H_22_N_2_)	5.0:1.0	1.55	Alocit 28.15

**Table 2 polymers-15-04290-t002:** Ion concentrations of seawater and tap water.

Ion	Seawater (mg/L)	Tap Water (mg/L)
Chloride (Cl^−^)	19,000	39.1
Sodium (Na^+^)	7500	86.2
Sulfate (SO_4_^2−^)	3300	58.8
Magnesium (Mg^2+^)	880	-
Calcium (Ca^2+^)	400	-
Potassium (K^+^)	490	-
Nitrate (NO_3−_)	-	16.1

**Table 3 polymers-15-04290-t003:** PH, salinity, and temperature of tap water and seawater.

Type of Water	pH		Salinity (%)		Temperature (°C)	
	Mean	SD	Mean	SD	Mean	SD
TW in the laboratory	7.15	1.35	0.05	-	20.70	0.58
SW in the laboratory	8.17	0.17	3.26	0.18	19.50	0.45
SW in the real sea environment	8.14	0.16	3.27	0.19	24.62	1.87

**Table 4 polymers-15-04290-t004:** Averaged coating thickness and bond strength values, with standard deviations.

Exposure Condition	Epoxy Type		E1				E2				E3			
	Exposure Period	Coating Device	T (µm)	S.D. (µm)	*f_bond_* (MPa)	S.D. (MPa)	T (µm)	S.D. (µm)	*f_bond_* (MPa)	S.D. (MPa)	T (µm)	S.D. (µm)	*f_bond_* (MPa)	S.D. (MPa)
Tap Water (TW)	7D	D1	27.78	5.17	0.87	0.32	162.22	15.71	2.27	0.29	146.67	26.25	3.04	0.26
		D2	82.89	4.94	1.89	0.34	310.00	58.94	2.29	0.27	291.11	38.62	1.64	0.37
		D3	42.22	4.23	0.91	0.17	128.89	3.14	1.81	0.22	304.44	20.43	1.75	0.23
	28D	D1	30.67	3.81	1.74	0.40	176.67	14.40	2.52	0.13	200.00	38.39	1.24	0.19
		D2	54.67	3.31	1.15	0.19	224.44	20.61	2.22	0.26	156.67	36.67	2.79	0.34
		D3	91.33	7.91	1.41	0.33	143.33	20.00	1.89	0.51	324.44	43.40	2.12	0.47
	56D	D1	30.00	2.37	1.16	0.20	313.33	125.11	3.02	0.12	166.67	9.81	2.33	0.08
		D2	61.33	22.65	1.39	0.20	237.78	26.43	2.75	0.24	187.78	29.10	2.14	0.65
		D3	123.33	26.73	0.87	0.29	140.00	19.63	1.95	0.67	313.33	47.69	1.67	0.17
	91D	D1	112.67	9.54	1.05	0.10	160.00	9.43	0.97	0.27	154.44	11.33	2.05	0.17
		D2	91.33	10.93	1.10	0.30	227.78	9.56	2.06	0.28	234.44	20.06	2.14	0.32
		D3	174.89	24.08	1.46	0.47	126.67	0.00	2.53	0.68	252.22	69.99	3.15	0.15
Sea Water (SW)	7D	D1	97.56	26.96	2.62	0.43	102.22	31.54	1.33	0.20	92.00	18.06	1.86	0.57
		D2	54.67	9.49	0.99	0.44	181.11	30.10	2.16	0.21	182.22	41.84	1.39	0.72
		D3	42.44	14.73	1.10	0.51	173.33	26.81	2.48	0.06	243.33	21.26	2.32	0.14
	28D	D1	241.11	12.57	2.61	0.26	142.22	35.52	1.40	0.13	263.33	57.35	1.32	0.39
		D2	115.56	11.00	2.25	0.42	222.22	40.40	2.70	0.37	100.00	11.86	0.45	0.12
		D3	51.56	6.31	0.74	0.05	151.11	32.81	0.98	0.24	164.44	60.63	1.30	0.65
	56D	D1	253.33	98.99	1.39	0.33	196.67	43.20	1.16	0.38	105.56	32.81	1.02	0.05
		D2	47.56	6.56	0.57	0.13	197.78	10.30	2.18	0.55	238.89	39.85	1.10	0.32
		D3	58.89	10.90	0.66	0.10	92.22	8.31	1.50	0.40	116.67	35.59	1.03	0.15
	91D	D1	62.22	10.42	0.98	0.17	183.33	18.86	2.25	0.25	100.22	8.02	2.74	0.20
		D2	65.78	12.54	1.16	0.33	160.00	15.15	2.49	0.62	171.11	29.98	2.34	0.53
		D3	81.11	11.70	1.71	0.56	131.67	11.67	2.01	0.31	118.89	9.56	1.98	0.76
Real Sea (RS)	7D	D1	183.33	62.78	1.00	0.02	294.44	131.38	2.20	0.46	Not measured			
		D2	236.67	67.71	0.74	0.24	214.44	10.30	1.34	0.41	372.22	21.83	2.20	0.05
		D3	230.00	82.78	1.14	0.26	425.56	64.54	0.90	0.22	167.78	51.74	1.31	0.16
	28D	D1	198.89	50.36	1.89	0.15	476.67	151.51	2.36	0.09	Not measured			
		D2	191.33	24.99	1.44	0.42	240.00	89.57	2.23	0.41	352.22	5.67	0.90	0.44
		D3	230.00	17.85	0.67	0.41	206.67	56.83	2.63	0.49	635.56	99.23	3.21	0.18
	56D	D1	93.33	11.86	0.49	0.12	431.11	15.71	2.22	0.93	348.89	36.24	1.94	0.85
		D2	164.44	31.54	1.61	0.19	285.56	32.47	1.85	0.21	233.33	20.55	2.40	0.81
		D3	276.56	19.12	0.64	0.21	198.89	20.61	1.15	0.32	287.78	39.09	2.37	0.26
	91D	D1	253.33	47.69	3.02	0.09	262.22	58.20	2.72	0.49	Not measured			
		D2	226.67	51.93	1.25	0.31	141.11	26.15	0.09	0.03	202.22	33.26	2.27	0.71
		D3	104.00	4.25	2.44	0.14	253.33	52.49	2.20	0.78	251.11	46.93	3.14	0.16

**Table 5 polymers-15-04290-t005:** ANOVA test results for mean thickness and strength values by epoxy type.

**Coating Thickness (µm)**						
**Groups**	**Count**	**Sum**		**Average**		**Variance**
E1	95	11,166.7		117.54		6979.596
E2	95	19,436.7		204.60		7972.383
E3	95	21,603.3		227.40		13,266.681
Source of variation	SS	df	MS	*F*	*p*-value	*F* crit.
Between groups	638,636.3	2	319,318.17	33.9476	6.1643 × 10^−14^	3.02778
Within groups	2,652,553.9	282	9406.22			
Total	3,291,190.3	284				
**Bond Strength (MPa)**						
**Groups**	**Count**	**Sum**		**Average**		**Variance**
E1	93	116.9		1.26		0.421
E2	93	179.8		1.93		0.591
E3	93	178.9		1.92		0.679
Source of variation	SS	df	MS	*F*	*p*-value	*F* crit.
Between groups	27.9	2	13.94	24.7317	1.32 × 10^−10^	9.52465
Within groups	155.5	276	0.56			
Total	183.4	278				

**Table 6 polymers-15-04290-t006:** ANOVA test results for the mean thickness and strength values by coating method.

**Coating Thickness (µm)**							
**Groups**	**Count**		**Sum**		**Average**		**Variance**
D1	90		16,406.0		182.29		15,184.540
D2	90		15,775.3		175.28		7009.231
D3	90		15,469.3		171.88		9557.819
Source of variation	SS		df	MS	*F*	*p*-value	*F* crit.
Between groups	5069.3		2	2534.67	0.2395	0.7872	3.02960
Within groups	2,825,891.4		267	10,583.86			
Total	2,830,960.8		269				
**Bond Strength (MPa)**							
**Groups**	**Count**		**Sum**		**Average**		**Variance**
D1	90		168.2		1.87		0.665
D2	90		152.7		1.70		0.622
D3	90		142.3		1.58		0.596
Source of variation	SS	df	MS		*F*	*p*-value	*F* crit.
Between groups	3.8	2	1.89		3.0090	0.0510	3.02960
Within groups	167.6	267	0.63				
Total	171.3	269					

## Data Availability

Available upon request to the corresponding author.

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
