# Peer review of "Interfacial Bond Properties of Underwater Concrete Coated with Bisphenol A Epoxy Resins"

_polymers, 2023, doi:10.3390/polym15214290_

Round 1
Reviewer 1 Report
Comments and Suggestions for Authors
1- Describe about the novelty of the paper.
2-add mechanical properties of material in abstract.
3-remove “the” in subscription of all figures.
4- Check the units of all tables.
5-whats the effect of exposure period on Coating thickness under TS condition
6-whats the effect of exposure period on thickness under TS condition
7-increase the quality of fig 1a.
8-describe about porosity
9- Whats the effect of coating thickness according to the type of epoxy.
Comments on the Quality of English Language
1- Describe about the novelty of the paper.
2-add mechanical properties of material in abstract.
3-remove “the” in subscription of all figures.
4- Check the units of all tables.
5-whats the effect of exposure period on Coating thickness under TS condition
6-whats the effect of exposure period on thickness under TS condition
7-increase the quality of fig 1a.
8-describe about porosity
9- Whats the effect of coating thickness according to the type of epoxy.
Author Response
We sincerely appreciate your valuable time and effort in reviewing this manuscript. Please check the attached file for the detailed responses.

Reviewer 2 Report
Comments and Suggestions for Authors
1. Details related to the size of Zinc particles and the amount (weight fraction, volume fraction) need to be provided since this will affect the bond properties and viscosity of the resin.
2.What was the pH and salinity of the sea water in the laboratory and in the "real sea environment"
3. What was the temperature of the aqueous solutions
4. Was any surface preparation conducted on the concrete. Were these first placed in water etc for a period of time prior to coating? If so what was the level of moisture penetration? In actual applications it is likely that the concrete will already be saturated with moisture prior to a coating being applied if this is a repair condition. The encapsulation of moisture can lead to severe and accelerated deterioration. How would this be assessed?
5. E2 and E3 both have additives added to the resin and these appear to affect both viscosity and bond strength. Can further clarifications be provided as related to the increase in viscosity over that of E1 and how that affected ease of placement and voids?
6. On line 259 on page 7 the authors mention "internal pores" Are these trapped between the concrete surface and resin, within the resin, elsewhere? What is the genesis of these? Were failures in pull-out seen to emanate from these?
7. The standard deviations of results in Figure 6 for example are of the same order of magnitude as the mean. How can the viability of data be assessed in such cases? The authors need to discuss this further not just for this set of Figures but others with the same concern
8. The presence of a weak relationship between strength and thickness suggest a weak interface as otherwise bond strength decreases with increases in adhesive thickness. Were any shear tests done on the resin systems themselves to measure shear strength and stiffness and compare them?
9. In Figure 8 it may be of value to compare performance to that of an exposed set of specimens (i.e without immersion). This would provide a better determination of overall effects and level of change.
10. While the authors do a good job in explaining statistical variation a slightly more nuanced set of analysis that looked at the relative effect of the different variables to highlight those that had the most effect and those that had the least would be illuminating both for the reader and the authors.
Comments on the Quality of English Language
Minor editing for typos is necessary
Author Response

(The authors gave the same response as above.)

Reviewer 3 Report
Comments and Suggestions for Authors
This manuscript investigates the interfacial bond properties of underwater concrete coated with bisphenol A epoxy resins. The authors examine the effects of epoxy type, coating method, exposure conditions, and exposure period on coating thickness and bond strength. The results show coating thickness increased in real sea conditions compared to tap water, while bond strength was mostly unaffected by conditions except for one epoxy type. The study provides valuable experimental data on factors influencing underwater coating performance. However, the conclusions drawn about relationships between thickness, bond strength and test variables seem preliminary given the high variability in results. More controlled testing may be needed to definitively determine which factors significantly impact bond strength and durability. Overall, this is a well-designed study generating useful data on underwater coatings, but conclusions could be strengthened by additional experimentation and analysis.
comments:
- In the introduction, please provide more background on the different types of epoxy resins tested and their potential advantages/disadvantages for underwater coating.
- the statement "This assumption is based on the notion that controlled environments can enhance interfacial bonding..." seems speculative. Please provide a reference or explanation for this assumption.
- please explain the high standard deviation in thickness values under real sea conditions. Is this expected or does it indicate issues with consistency of application method?
- please provide p-values and test statistics for the ANOVA results in Table 4 to support statements about statistical significance.
- please explain why the conventional roller exhibited the highest thickness variation, contrary to expectations. Is this a limitation of the coating application methods?
- please discuss why exposure conditions significantly affected coating thickness but not bond strength. Is there an explanation for this discrepancy?
- please explain the decreasing thickness trend under real sea conditions. Is this indicative of degradation over time?
- please explain why E3 showed increased bond strength over time while other epoxies did not. Is this a key finding regarding epoxy selection?
- In the discussion section, please comment on the high variability in results and how this affects drawing definitive conclusions. What are limitations?
- In the discussion, please expand on how these results will inform optimal application methods and material selection for underwater coatings.
- Please discuss variability between samples with the same epoxy/method/condition - are results repeatable? How many samples showed each trend?
- Please comment on whether the exposure periods up to 91 days are sufficient to evaluate coating durability long-term.
- Please discuss differences in performance between epoxies in more detail, especially E3 bond strength increasing over time.
- In conclusions section, avoid definitive statements about factors influencing results given high variability. Soften conclusions appropriately.
Minor editing of English language required.
Author Response

(The authors gave the same response as above.)

Round 2
Reviewer 3 Report
Comments and Suggestions for Authors
Suitable for publication